



# Root growth, water uptake, and sap flow of winter wheat in response to different soil water availability

Gaochao Cai[1], Jan Vanderborght[1], Matthias Langensiepen[2], Andrea Schnepf[1], Hubert Hüging[2], and Harry Vereecken[1]

[1] Agrosphere, Institute of Bio- and Geosciences (IBG-3), Forschungszentrum Jülich GmbH, 52428, Jülich, Germany

[2] Institute of Crop Science and Resource Conservation, Faculty of Agriculture, University of Bonn, Katzenburgweg 5, 53115, Bonn, Germany

*Correspondence to*: Gaochao Cai (g.cai@fz-juelich.de)

**Abstract.** How much and where water is taken up by roots from the soil profile are important questions that need to be answered to close the soil water balance equation and to describe water fluxes in the soil-plant-atmosphere system. Physically-based root water uptake (RWU) models that relate RWU to transpiration, root density, and water potential distributions have been developed but far less used or tested. This study aims at evaluating the simulated RWU of winter wheat by the empirical Feddes-Jarvis (FJ) model and the physically-based Couvreur (C) model for different soil water conditions and soil textures against sap flow measurements. Soil water content (SWC), water potential, and root development were monitored non-invasively at six soil depths in two rhizotron facilities that were constructed in two soil textures: stony vs. silty with each three water treatments: sheltered, rainfed, and irrigated. Soil and root parameters of the two models were derived from inverse modeling and simulated RWU was compared with sap flow measurements for validation. The different soil types and water treatments resulted in different crop biomass, root densities and root distributions with depth. The two models simulated the lowest RWU in the sheltered plot of the stony soil where RWU was also lower than the potential RWU. In the silty soil, simulated RWU was equal to the potential uptake for all treatments. The variation of simulated RWU among the different plots agreed well with measured sap flow but the C model predicted the ratios of the transpiration fluxes in the two soil types slightly better than the FJ model. The root hydraulic parameters of the C model could be constrained by the field data but not the water stress parameters of the FJ model. This was attributed to differences in root densities between the different soils and treatments which are accounted for by the C model whereas the FJ model only considers normalized root densities. The impact of differences in root density on RWU could be accounted for directly by the physically-based RWU model but not by empirical models that use normalized root density functions.

## 1 Introduction

Root water uptake (RWU) is a vital process for plant functioning since it conditions nutrient transport and balances transpiration. Estimating RWU is needed to make predictions of crop water use, to assess water and nutrient use efficiency in function of root architecture and environmental controls, and to design efficient water and nutrient resources management in agricultural practices (Molz, 1981). However, quantifying RWU for water and nutrient management in different regions and climates continues to be a challenge due to the lack of knowledge of key RWU parameters and appropriate description of the RWU process (Vereecken et al., 2016). Typically, RWU is estimated from the transpiration demand, which is calculated from the canopy energy balance under the assumption that the crop is well-watered. Different soil water balance models have been developed that allow estimating RWU using different parameterizations of the root system and water uptake mechanisms. However, the availability of field plot scale experiments in different soil textures and for different soil water regimes that are needed to validate and improve these models is very limited.





In many soil water balance models that are used to predict RWU Richards equation is used for calculating water flow in unsaturated soils and a sink term is defined that describes RWU:

$$\frac{\partial \theta}{\partial t} = \nabla(K(h)\nabla(h + z)) - S \qquad (1)$$

where $\theta$ represents the volumetric soil water content (SWC) [$L^3 L^{-3}$], $t$ time [T], $K$ the soil hydraulic conductivity [$L T^{-1}$], $h$ the

soil water pressure head (SWP) [L], $z$ the elevation [L], and $S$ the sink term [$L^3 L^{-3} T^{-1}$] defined as the volume of water removed from a unit volume of soil due to root extraction. A popular macroscopic RWU model that has been used to quantify the sink term is the Feddes model (Feddes et al., 1976) because of its simplicity and low data requirement (Skaggs et al., 2006; Luo et al., 2003; Peters et al., 2017). It uses the normalized root length density distribution and stress functions to determine the distribution of the sink term in the root zone. Piecewise linear stress functions define how the sink term at one location in the

root zone is reduced as a function of the SWP and this function depends in turn on the potential transpiration rate. This model was refined later to the Feddes-Jarvis model by adding a factor to account for increased water uptake, i.e. uptake compensation, from moister soil layers when uptake from drier layers is reduced (Jarvis, 1989; Šimůnek and Hopmans, 2009).

Besides transpirational demand and soil water pressure head (SWP), RWU is also influenced by root hydraulic properties (i.e.

root hydraulic conductance) which may vary over time due to root development and growth (Doussan et al., 1998; Steudle, 2000; Javaux et al., 2008). Root hydraulic properties determine the resistance to water flow within the plant and define the water potential losses along the sap flow from the roots to the shoot and the leaves (Bechmann et al., 2014). The relation between soil water and leaf water potentials, and sap flow depends on root hydraulic properties which therefore should be considered in RWU models (Vereecken et al., 2015; Vadez, 2014). Physically-based macroscopic RWU models were

developed that describe water fluxes in the soil-root (or soil-root-plant) system based on water potentials and conductivities or conductances of the soil and the root system. Nimah and Hanks (1973) characterized water uptake as a function of root density, axial root conductance, and water potential at the root collar. Heinen (2001) considered root hydraulic properties and water pressure head at the root-soil interface in the RWU model but without considering water uptake compensation. van Lier et al. (2008) developed a 1-D water flow model in which RWU rate was a function of root surface water potential and root radius.

This model considered implicitly lateral flow from soil to root with implicit compensation mechanism but did not include the information of axial root hydraulic conductances.

In order to present a mechanistic description of the RWU process that contains physically defined parameters, Couvreur et al. (2012) developed a 3-D model based on the approach of root system hydraulic architecture (Doussan et al., 2006; Javaux et

al., 2008). In this model, RWU is dependent on root system hydraulic conductance ($K_{rs}$), the root distribution, and the difference between the local soil water potential and the water potential at the root collar. Variations of this potential difference with depth in the root zone lead to water uptake compensation. For crops with small lateral variations in root length density (RLD), this 3-D model could be upscaled to a 1-D model (Couvreur et al., 2014) which shows similarities to the models of Nimah and Hanks (1973) and of Amenu and Kumar (2008). Cai et al. (2017) obtained the root hydraulic parameters of the 1-D upscaled

Couvreur model for winter wheat (Couvreur et al., 2014) by inverse modeling using time series of soil water potential, water content, and root length density measurements in the field. Since the upscaled root hydraulic parameters have physical meaning, the upscaled parameters obtained from inverse modeling could be compared to and were found to be consistent with parameters that were derived from direct measurements of hydraulic properties of root segments and models of the hydraulic root architecture (Couvreur et al., 2014).




Another way to validate the inversely estimated parameters is to evaluate whether the model is able to predict the RWU and its reduction when SWP decreases. For crops with a small water capacity, the RWU corresponds closely with the transpiration rate. Measurements of crop transpiration can therefore be used to parameterize or validate RWU models.

Many techniques have been used to investigate transpiration ranging between the single plant and catchment scale (Twine et al., 2000; Allen et al., 1989; Jaeger and Kessler, 1997). At the field plot scale, weighing lysimeters allow to measure transpiration (e.g., Groh et al., 2016; Garré et al., 2011). A disadvantage of lysimeters is that they are costly and, although possible (e.g., Garré et al., 2011; Vandoorne et al., 2012), root distributions are difficult to measure in lysimeters and their spatial growth is influenced by the confined soil space which also frequently causes undesired boundary effects (e.g. high root

length densities at lysimeter walls). Measuring sap flow with the thermoelectric method is a direct and *in situ* technique which was discovered by Huber (1932). It was used to estimate transpiration for different trees species (Granier et al., 1996; Cermak et al., 2004; Massai and Remorini, 2000) and crops (Chabot et al., 2005; Langensiepen et al., 2014; Cohen et al., 1993). Due to limitations of sensor installation on small and vulnerable crop stems, sap flow measurements on crops with small stem diameters of less than 5 mm are practically challenging. Senock et al. (1996) provided first sap flow measurements for wheat

under field conditions but the values were within 10 % of gravimetric measurements and the experimental verification of a high sap flow rate (up to 5 g h$^{-1}$) is not available. Applying an empirical method for calculating sap flow from standard stem heat sensor outputs, Langensiepen et al. (2014) obtained close agreements between measured sap flow and transpiration rates measured with a standard eddy covariance system. Continuous sap flow measurements can be carried out with modern logging techniques (e.g. multiplexer and data logger), providing insight into the temporal dynamics of transpiration and how for

instance RWU changes when SWP decreases over time. The use of sap flow measurements to validate theories of RWU was demonstrated for trees (Gong et al., 2006; Howard et al., 1996; Green and Clothier, 1998), but for crops, in particular wheat, such a validation has not yet been performed under field conditions.

The main objective of this study is to investigate whether a physically-based model for RWU can simulate the effect of different

soil water availability on wheat RWU resulting from differences in soil water application and differences in soil water retention characteristics. This includes testing whether parameters of such a model can be calibrated using measurements of soil water content, water potential, and root density and validating the calibrated model against sap flow measurements. Second, we investigated whether differences in crop shoot and root developments between treatments with different soil water availability lead to different model parameter estimates and whether these parameters estimates can be linked to directly observable

properties of the root system.

Therefore, water potentials and contents, root distributions, crop development and sap flow were monitored in six plots (two soil types and three water application treatments) and used to parameterize two RWU models: the empirical Feddes-Jarvis model (FJ model) and the physically-based Couvreur model (C model).

**2 Materials and methods**

The experimental set up of the plots was described in detail in Cai et al. (2016) and the model setup and the inverse modeling procedure that was used to determine the parameters by Cai et al. (2017). For more detailed information on the setup and the inverse modeling procedure, we refer the reader to these publications.





### 2.1 Setup of the test site

Two instrumented rhizotron facilities were constructed in the upslope and the downslope of a cropped field in Selhausen (Germany, 50 ̊52'N, 6 ̊27'E). The field is on a west-facing slope (smaller than 4º) and characterized by a high stone content (up to 60 %) in the upslope and silty texture in the downslope. Each facility was divided into three plots of 7 m length $\times$ 3.25 m width. To produce a gradient in soil water availability, one plot was sheltered from rain, one plot was rainfed, and one plot was irrigated by drip-irrigation. A sketch of the facilities with the location of the sheltered, rainfed and irrigated plots and the wooden framed trenches is shown in Fig. 1.

Precipitation and other meteorological data for the calculation of reference evapotranspiration ($ET_o$) using the FAO56 Penman-Monteith equation (Allen et al., 1998) were obtained from a weather station located in close proximity to the two facilities. The average annual precipitation for the past 50 years in this area was 699 mm (Knaps, 2016).

Winter wheat (variety Ambello) was sown at a density of 300 – 320 seeds m$^{-2}$ on 31 Oct. 2013 in all plots and harvested on 17 July 2014 in the stony soil (upper facility) and on 31 July 2014 in the silty soil (lower facility) as the contrasting soil-water regimes in the two soils affected ripening times. Total shoot biomass was harvested in an area of 7.31 m$^2$ (3.25 m $\times$ 2.25 m) in each plot and weighed after oven drying. Leaf area index (LAI) was measured using a plant canopy analyzer (LAI-2200, LI-COR, Inc. USA) and ranged from 0.8 to 2.5 in the stony soil and from 0.8 to 4.0 in the silty soil between 8 Apr. and 14 July 2014 (Fig. 2a). Precipitation depth between the seeding and harvest was 434.49 mm for the stony and 495.89 mm for the silty soil, the difference resulting from different growth period lengths in both facilities. Fig. 2b shows the cumulative amount of water received by the three plots in both soils.

### 2.2 Measurements of soil moisture, root distribution, and sap flow

*Soil water content and potential*
Time domain reflectometers (TDR), tensiometers (T4e, UMS GmbH, München, Germany), and matrix water potential sensors (MPS-2, Decagon Devices Inc., UMS GmbH München, Germany) were installed in each plot at 0.1, 0.2, 0.4, 0.6, 0.8, and 1.2 m depth in the vertical walls of the facilities to monitor hourly SWC and soil water potential.

*Root observation*
Root distributions were measured non-destructively at weekly intervals from 11 Feb. 2014 to 11 July 2014 in the stony soil and from 14 Mar. to 24 July 2014 in the silty soil with a minirhizotron camera (Bartz Technology Corporation, Carpinteria, CA, USA) in 7-m-long horizontally installed rhizotubes. Three replicates of rhizotubes were installed at the same depths as the soil moisture sensors. Root images with the size of 16.5 mm $\times$ 23.5 mm were taken from left and right sides at 20 fixed locations along each tube and were analyzed subsequently using the software Rootfly (Wells and Birchfield, 2009) to determine the length of roots per area of the image. Root length densities were therefore expressed in units of length per surface. To calculate the total root length below a unit surface area, both root length and root counts per image surface were considered in Cai et al. (2017). We assumed that root lengths in the images were proportional to root counts. Using root counts has the advantage of avoiding to use the empirical soil thickness (e.g. 2 mm) viewed by the camera in the estimation of absolute total root length. The root counts were associated with a soil volume that corresponds with the diameter (height, 64 mm) and the radius (width, 32 mm) of the tube, and the image width (depth, 16.5 mm) to obtain an estimate of root length density (Cai et al., 2017). The root densities were subsequently integrated over depth to obtain the total root length below a unit surface area.

*Sap flow*



Sap flow was determined with SGA3 Dynagage sap flow sensors (Dynamax Inc., Houston, USA) in five randomly selected wheat tillers located in the center of each plot. They were continuously operated from 23 May 2014 to 6 July 2014. Signals of the sap-flow sensors were scanned every 60 seconds with Dynamax control units consisting of voltage regulators, AM 16/32B multiplexers and CR1000 dataloggers (Dynamax Inc., Houston, USA; Campbell Scientific, Logan, Utah). The readings were

5 averaged every 10 minutes, stored in a text file and processed with an R script containing the standard calculation procedures for computing sap-flow from Dynagage files (Dynamax, 2009) and an improved post-processing method for removing the noise from standard calculations (Langensiepen et al., 2014). Tiller density was determined in a fixed area of 1 m$^2$ for each plot and used for converting average sap flow rate (g d$^{-1}$ tiller$^{-1}$) to an area-based transpiration rate (cm d$^{-1}$).

**2.3 Root water uptake models and parameterizations**

*Model description*

We used two 1-D RWU models: the FJ model (Šimůnek and Hopmans, 2009) and the physically-based C model (Couvreur et al., 2012; Couvreur et al., 2014), both of which considered water uptake compensation. The two models have been implemented in Hydrus-1D (Simunek et al., 2016). The sink term in the two models is calculated with following equations:

$$S_{FJ}(z) = T_{pot}\,\alpha_F(h)\,\text{NRLD}(z)\gamma \qquad (2)$$

$$S_C(z) = \min(T_{pot}, T_{act})\text{NRLD}(z) + K_{comp}(H(z) - H_e)\text{NRLD}(z) \qquad (3)$$

where $S_{FJ}$ and $S_C$ are the sink terms accounting for RWU rates in, respectively, the FJ and C models [L$^3$L$^{-3}$T$^{-1}$], $z$ elevation [L], $T_{pot}$ and $T_{act}$ the potential transpiration and transpiration under water stress condition [LT$^{-1}$], $\alpha_F$ the water stress function described below [-], $h$ the measured soil water pressure head (SWP) [L], NRLD the normalized root length density [L$^{-1}$], $\gamma$ the water uptake compensatory factor [-], $K_{comp}$ the compensatory RWU conductance of the root system [T$^{-1}$], $H$ the total hydraulic

head (sum of pressure head and elevation head) [L], and $H_e$ the effective root zone hydraulic head [L]. $H_e$ is the integration of soil hydraulic head and root distribution along the rooting depth ($l_z$, [L$^{-1}$]) profile:

$$H_e = \int_0^{l_z} H(z)\text{NRLD}(z)\, dz \qquad (4)$$

$T_{pot}$ is given by:

$$T_{pot} = ET_o K_c\,(1 - e^{-k*LAI}) \qquad (5)$$

where $K_c$ is the crop coefficient [-] that accounts for changes in evapotranspiration with crop development (Allen et al., 1998) (Table 1), LAI the leaf area index [-], and $k$ the light extinction coefficient (0.6 was used (Xin-Yang and Yang-Ren, 2013; Rodriguez et al., 2001; Oroud, 2012; Bingham et al., 2009)). In the C model, the leaf water hydraulic head, $H_{leaf}$ [L] is related to $T_{pot}$, the equivalent root system hydraulic conductance, $K_{rs}$ [T$^{-1}$], and the effective root zone hydraulic head, $H_e$ [L], by:

$$T_{pot} = K_{rs}(H_e - H_{leaf}) \qquad (6)$$

as long as $H_{leaf}$ [L] is larger than a critical leaf hydraulic head, $H_{leaf\_crit}$ (-16000 cm was used in this study (Wesseling, 1991)). When the leaf water potential equals $H_{leaf\_crit}$, the transpiration rate is reduced and the actual transpiration rate $T_{act}$ is obtained from:

$$T_{act} = K_{rs}(H_e - H_{leaf\_crit}) \qquad (7)$$

The parameters $K_{rs}$ and $K_{comp}$ [T$^{-1}$] of the C model depend on the root development since the root system conductance increases

when the root system grows. In line with Cai et al. (2017), we assumed that $K_{rs}$ and $K_{comp}$ were proportional to the total root length of the root system that is derived from the integral of the RLD over depth.

For the FJ model, the RWU under water stress condition was constrained by a piecewise function ($\alpha_F$) that is dependent on SWP:




$$\alpha_F(h) = \begin{cases} 0 & h \notin [h4, h1] \\ \frac{h - h_1}{h_2 - h_1} & h \in (h2, h1] \\ 1 & h \in [h3, h2] \\ \frac{h - h_4}{h_3 - h_4} & h \in [h4, h3] \end{cases} \qquad (8)$$

where $h_{1,4}$ and $h_{2,3}$ are the thresholds of SWP where RWU is completely constrained ($S = 0$), and arrives the maximum, respectively. The value of $h_3$ is a function of $T_{pot}$ (Brandyk and Wesseling, 1985):

$$h_3 = \begin{cases} h_{3l} & T_{pot} \in [0, T_{3l}] \\ h_{3h} + \frac{(h_{3l} - h_{3h})(T_{3h} - T_{pot})}{(T_{3h} - T_{3l})} & T_{pot} \in (T_{3l}, T_{3h}) \\ h_{3h} & else \end{cases} \qquad (9)$$

where $T_{3h}$ and $T_{3l}$ were set to 0.02 cm h$^{-1}$ and 0.004 cm h$^{-1}$ (Yang et al., 2009).

The water uptake compensation in the C model is described by the second term on the right-hand side of Eq. 3. For the FJ model it is controlled by an empirical factor ($\gamma$) that is water stress related:

$\gamma = 1/\max(\omega, \omega_c)$ (10)

$\omega = \int_{l_z} \alpha_F(h) \, NRLD(z) \, dz$ (11)

where $\omega_c$ is the critical water stress threshold [-] which ranges between 0 and 1 corresponding to, respectively, full compensation and no-compensation (Šimůnek and Hopmans, 2009; Jarvis, 1989).

The soil hydraulic properties were described by the combined Mualem-van Genuchten equations (Mualem, 1976; Van

Genuchten, 1980):

$$\theta(h) = \begin{cases} \theta_r + \frac{\theta_s - \theta_r}{[1 + |\alpha h|^n]^m} & h \in (-\infty, 0) \\ \theta_s & else \end{cases} \qquad (12)$$

$K(S_e) = K_s S_e^l [1 - (1 - S_e^{1/m})^m]^2$ (13)

where $\theta_r$ and $\theta_s$ are the residual and saturated water content [L$^3$L$^{-3}$], $\alpha$ [L$^{-1}$], $n$ ($n > 1$), $m$ ($m = 1 - 1/n$), and $l$ are shape parameters, $K$ and $K_s$ are the unsaturated and saturated hydraulic conductivity [LT$^{-1}$] , $S_e$ is the effective saturation [-]: ($\theta - \theta_r$)/($\theta_s - \theta_r$).

*Inverse Modeling and model setup*

The parameters of the van Genuchten (1980) soil water retention function were fitted using measured SWC and soil water head data (Cai et al., 2016) (Table 2). The parameters, $K_s$, $l$, $h_{3h}$, $h_{3l}$, and $\omega_c$ of the FJ model, $K_{rs}$, and $K_{comp}$ of the C model were inversely estimated by fitting simulated to hourly measured SWP and SWC. For the stony soil, a time series from 11 Feb. to

14 July 2014 and for the silty soil from 22 May to 30 July 2014 was used. Besides the time series of the SWP and SWP, also other variables that were derived from these time series, such as changes in SWP and SWC over time and water storage in the soil profile were included in the objective function that was minimized in the fitting procedure as described in Cai et al. (2017) and we refer for the details to that paper. In the current study, observations and simulations of soil moisture dynamics for the three treatments per soil type (i.e. stony and silty soils) were lumped into one objective function whereas Cai et al. (2017) used

only data from the sheltered plot in the stony soil. Hence, for each soil type, the same soil and RWU parameters were used to simulate RWU for the three treatments. But, since the two objective functions with data from the two different soil types were optimized independently, different soil and RWU parameters were obtained for the two different soils.

The 1-D Richards equation was solved numerically using Hydrus in a 145 cm deep soil profile for the stony soil and a 300 cm

deep profile for the silty soil using a spatial discretization of 1 cm. Two soil layers with different hydraulic properties, the topsoil (0 – 30 cm) and the subsoil (30 – 145 cm for the stony soil and 30 – 300 cm for the silty soil), were considered at both




facilities. An atmospheric boundary condition was used at the top and a free drainage boundary condition at the bottom (Simunek et al., 2013). The soil water pressure heads measured at the start of the simulation period were used as initial conditions. In order to consider the root development during the growing period, the simulation period was split up into one-week periods during which a constant RLD profile was assumed. The parameters $K_{rs}$ and $K_{comp}$ of the C model, which were

assumed to depend on the total root length, were hence adjusted at weekly intervals. One set of parameters: $K_{rs\_ini}$ and $K_{comp\_ini}$ that correspond with the RWU parameters from the sheltered plot during the first week of the simulation period were estimated using inverse modeling. $K_{rs}$ and $K_{comp}$ during the $i$th week of a certain water treatment were obtained by scaling $K_{rs\_ini}$ and $K_{comp\_ini}$ with the ratio of the integrated root length (integration of RLD over the soil profile) in week $i$ in that water treatment to the integrated root length during the first week in the sheltered plot. The initial conditions of a one-week period were derived

from the simulated SWP profile at the end of the previous one-week simulation period.

The model results were evaluated in terms of root mean square error (RMSE), mean bias error (ME), and an index of agreement ($d$):

$$\text{RMSE} = \sqrt{[\textstyle\sum_{i=1}^{N}(Sim_i - Obs_i)^2]/N} \tag{14}$$

$$\text{ME} = [\textstyle\sum_{i=1}^{N}(Sim_i - Obs_i)]/N \tag{15}$$

$$d = 1 - [\textstyle\sum_{i=1}^{N}(Sim_i - Obs_i)^2]/[\textstyle\sum_{i=1}^{N}(|Sim_i - \overline{Obs}| + |Obs_i - \overline{Obs}|)^2] \tag{16}$$

where $Sim$ and $Obs$ are simulated and measured variables, $i$ is the index of a given variable, and N the number of observations.

## 3 Results and discussion

We first discuss the effect of water treatments and soil textures on crop and root development. In the second part, we discuss

the inverse estimation of RWU parameters of the FJ and C models from measured SWC and SWP. In the third part, simulated RWU by the two models in the different soils and water treatments are discussed and compared with sap flow measurements. In the last part, we discuss a sensitivity analysis that was carried out to evaluate the effect of the different development of the wheat crop in the different soils and water treatments on the simulated water uptake.

### 3.1 Effect of water treatment on crop and root development

Tiller densities and crop biomass in the three different water treatments in the two soils are shown in Table 3. Contrasting soil water availability affected crop biomass growth and yield. Less water application (sheltered plot received 55.13 % and 44.52 % of the water received by the irrigated plots in the stony and silty soil, respectively, Figure 2b) reduced the tiller density in the sheltered plot with respect to the irrigated plot by 38.4 % in the stony and 11.3 % in the silty plots, and reduced the biomass by 58.8 % in the stony and 40.8 % in the silty plots. The biomass of wheat in the treatments that received less water was

reduced stronger than the tiller density as was also reported by Musick and Dusek (1980). The tiller density and biomass were generally higher in the silty than in stony soil, especially for the sheltered plots. The higher water holding capacity of the silty soil supplying more available water in the subsoil for root extraction may account for the difference.

As for the belowground part of the crops, RLD decreased gradually downwards for all plots of the two facilities at the beginning

of the measurements (Fig. 3). The RLD in the shallow layers (-10 to -20 cm) was similar in the stony and silty soils, ranging from 0.12 to 0.67 cm cm$^{-2}$. However, larger differences in RLD between the two soils were observed at greater depths (-60 to -120 cm depth). In the stony soil, maximal root densities were observed at shallower depths (-40 cm in the sheltered and irrigated plots and -60 cm in the rainfed plot) than in the silty soil (-60 cm in the irrigated and -80 cm in the sheltered and rainfed plots). Furthermore, the maximal root length densities were considerably higher in the silty than in the stony soil (note





the difference in color scale). The root density distributions showing maximal densities at greater depths are markedly different from the root density profiles that have been observed for winter wheat using soil coring in loamy soil (Zhang et al., 2004) and in soils with seven different textures (from clay to sandy loam) (e.g., White et al., 2015; Zhang et al., 2004). This might on the one hand be due to a great amount of water stored at those depths in the silty soil but probably also nutrient distribution in the

soil profile at this site, which might have promoted root development in deeper soil layers (Thorup-Kristensen et al., 2009; Yang et al., 2003). On the other hand, some studies indicated that root length densities estimated from rhizotubes may underestimate the root densities in surface soil layers due to temperature effects (Fitter et al., 1998), or roots growing parallel to the horizontal plane not intersecting the tube surface (Meyer and Barrs, 1991). We obtained root lengths ranging from 1.5 to 7.0 km m$^{-2}$ which is within the range of the results from White et al. (2015) who investigated root development of 11 winter

wheat varieties in four different soils (from clay to sandy loam) in the UK. They found an average of 9.8 km m$^{-2}$ from the samples to 1 m depth. The lower estimate might be due to an underestimation of the root density in the upper 30 cm using the rhizotubes.

Root senescence was observed at the end of the growing season. It started in the upper soil layers and progressively moved to

deeper layers, which is more obvious in the three plots of the stony soil after 21 May. Furthermore, root senescence in shallower layers (above 30 cm) occurred simultaneously with root development in deeper layers (below 30 cm).

The observed root development in the two different soils and for the different water treatments show opposite reactions to soil water availability. On the one hand, lower water availability in the stony soil led to a lower root density and lower total root

length than in the silty soil (Table 3). The same behavior was observed when comparing the sheltered with the rainfed and irrigated plots in the stony soil. In the silty soil, however, an increase in root density was observed when water availability decreased. When plants experience water deficits, above shoot development is reduced by different mechanisms (e.g. reduced leaf expansion by lower turgor, enhanced respiration, stomatal closure, and reduced photosynthesis) (Bunce, 1978; Wesselius and Brouwer, 1972; Mansfield and Atkinson, 1990). The reduction in shoot growth can be counteracted with an increase in

carbon allocation to the root zone as was shown in a review by Poorter et al. (2012) on environmental effects on biomass allocation. The ratio of total root length to aboveground biomass (Fig. 4) suggests that indeed a larger fraction of carbon was allocated to the roots in the sheltered than in the rainfed or irrigated plots both in the stony and silty soils. Although the differences in the ratio between the two soils are not so large, the total root length per kg shoot biomass was larger in the silty than in the stony soil. This seems at first sight contradictory to the lower water (and nutrient) availability in the stony than in

the silty soil. This might reflect that other factors like soil mechanical strength may have stimulated root growth more in the silty than in the stony soil (Unger and Kaspar, 1994; Merotto Jr and Mundstock, 1999).

### 3.2 Inverse estimation of soil and root water uptake parameters of the Feddes-Jarvis and Couvreur models from soil water contents and water potential measurements

Time series of observed and simulated SWC and SWP are illustrated in Fig. 5 and 6 for the plots with different water treatments

of the stony and silty soils, respectively. As expected, the irrigated plots were wetter than the rainfed and sheltered plots but in the top layers of the silty soil measured water contents and pressure heads decreased between irrigation events to similar low values as in the non-irrigated plots. For the period that measurements were carried out in both soils (from mid of May until beginning of July) the SWPs in the sheltered and rainfed plots were more negative in the stony than in the silty soil suggesting that the crop experienced more water stress in the stony soil. In both soils, the top layer dried out considerably and

low SWP (-10$^4$ cm) was reached as a result of high evaporation and transpiration demand. In the sheltered and rainfed plots of the stony soil, such low SWP were also reached in the deeper soil layers (-60 and -80 cm) whereas SWPs stayed higher at those depths in the silty soil due to the larger water holding capacity of the silty soil.





The statistics RMSE, ME, and $d$ of the SWC and SWP simulated by the two models are listed in Table S1. Since the statistics were very similar for both models, there was no notable difference between simulation accuracies of the FJ and C models. The values of RMSE for SWC in the stony soil (0.02 to 0.03 cm$^3$ cm$^{-3}$) were almost half of those in the silty soil whereas for SWP the values did not differ much between the two soils (from 0.3 to 0.9 log$_{10}$([-cm])). The larger RMSE of SWC in the silty soil is also due to the larger uncertainty in the measured SWC due to the variability of SWC between the four replicate TDR sensors (standard error of the sample mean reached 0.035 cm$^3$ cm$^{-3}$) (Cai et al., 2016).

The obtained soil hydraulic parameters, parameters of the water stress function of the FJ model, and root-system parameters of the C model are listed in Table 4. The corresponding hydraulic conductivity curves are plotted in Fig. S1. For the stony soil, the soil hydraulic parameters estimated by the two models were comparable but larger differences between the model parameters were obtained for the subsoil layer of the silty soil. Smaller (even negative) tortuosity parameters $l$ were obtained for silty than for the stony soil which implies that in the latter the hydraulic conductivities decrease stronger with a decrease in saturation degree (Eq. 13). For the same water content, hydraulic conductivities were higher in the stony than in the silty soil.

For the FJ model, parameters of the stress function were similar for the stony and silty plots, which implies that the estimated parameters were not sensitive to the different root density in the two different soils. It is important to note that the difference in root density between the different water treatments in one soil was not considered in the model since only one parameter set was used to simulate the different water treatments. The obtained threshold values of the stress function $\alpha_F$, $h_{3l}$ and $h_{3h}$ in Eq. 8 were higher than the lowest SWPs measured and simulated in the top- and subsoil layer in the sheltered and rainfed plots of the stony soil. Consequently, the FJ model simulated a reduction in RWU due to reduced water availability in these plots (Fig. 5b). For the silty soil, $h_{3l}$ and $h_{3h}$ were also higher than the lowest SWPs measured in the topsoil layer but lower than the SWPs in the subsoil. However, despite the lower SWPs in the topsoil and the low compensatory uptake (high $\omega_c$), no reduction in transpiration rate was simulated in the silty soil (Fig. 6b) as compared to the calculated potential transpiration rate. A first explanation for this observation is, that the high root density in the subsoil made that most of the water was simulated to be taken from the subsoil where SWP was high. Therefore, a reduction of uptake in the top layer where root densities were low would not affect the total uptake considerably and would require only a small compensatory uptake from the subsoil. Albasha et al. (2015) noted that compensatory water uptake could also be caused by increased root growth in soil layers where more water is available. The second explanation is, that the simulated SWP in the topsoil layer remained higher than corresponding measured values which is another reason why no reduction in transpiration was simulated in the silty soil.

Temporal changes in root system hydraulic conductance $K_{rs}$ of the C model is illustrated in Fig. 7 for the stony and silty soils. The $K_{rs}$ values in the different plots of the same soil were calculated using the same fitted initial $K_{rs\_ini}$ and RLDs so that the difference in $K_{rs}$ between the different plots of the same soil reflected differences in RLD. However, $K_{rs\_ini}$ was fitted independently for the two different soil types. The higher $K_{rs}$ obtained for the silty soil with the higher root density than the stony soil supports our hypothesis that the root system hydraulic conductance increases with the RLD. Considering the root system conductance that was normalized by the root length per soil surface area, the normalized root conductance was different for the two different soils. The value of the normalized $K_{rs\_ini}$ was 1.4 times larger and normalized $K_{comp\_ini}$ 8.2 times larger in the stony than in the silty soil. This indicated that for a single root segment the root conductance and compensatory ability was higher in the stony soil than in the silty soil. This difference does not support our assumption that $K_{rs}$ is directly proportional



to the RLD. It indicates that the different development of the root system in the stony soil, in which more water stress occurred, had an impact on the root hydraulic conductance of individual root segments.

To evaluate the uniqueness of the estimated parameters of the FJ and C models, response surfaces of the objective function were plotted. Selected contour plots in Fig. S2 show that the soil hydraulic parameters were identifiable. The parameters in the C model were also identifiable in both soils but $K_{rs}$ and $K_{comp}$ in the silty soil could be less precisely identified than in the stony soil. When RWU is not reduced and remains equal to the potential transpiration, which was the case in the silty soil (see later), Eq. 6 states that $K_{rs}$ can decrease without changing the RWU by decreasing $H_{leaf}$ until the $H_{leaf}$ reaches the critical leaf water potential. This explains why regions with low objective function values are bound by minimally possible $K_{rs}$ values but not by maximally possible $K_{rs}$ values in the silty soil. When RWU is lower than the potential transpiration, there is also a maximally possible $K_{rs}$ value so that leaf water potentials still reach the critical leaf water potential during the simulation period. In agreement with what was found by Cai et al. (2017), the response surface did not show a distinct global minimum for the water stress parameters in the FJ model.

In contrast to the current study, Cai et al. (2017) inversely estimated the soil hydraulic parameters and parameters of the FJ and C models using only observations from the sheltered plot in the stony soil. Inclusion of data from the rainfed and irrigated plots had an impact on the optimized soil hydraulic parameters (see values in parentheses in Table 4) whereas similar values of the root hydraulic conductances $K_{rs\_ini}$ and $K_{comp\_ini}$ were obtained. Including data that represent the hydraulic behavior of the soil under wetter conditions led to higher estimates of the hydraulic conductivity of the subsoil under wet conditions but lower estimates of hydraulic conductivities in the topsoil and in the subsoil for drier conditions (see Fig. S1). Using the parameter set obtained by Cai et al. (2017) resulted into a slightly better (e.g. for SWC, RMSE was 0.0057 and 0.0036 smaller for FJ and C models, and $d$ was 0.0257 and 0.0129 higher for FJ and C models) estimates of SWC and SWP in the sheltered plot but to an underestimation of the SWC and SWP in the rainfed and especially in the irrigated plot (see Fig. S3). This illustrates that soil hydraulic parameters that were obtained for a certain set of boundary conditions are not always transferable to other conditions. Combining experimental datasets that represent a wider range of boundary conditions is therefore preferable.

### 3.3 Simulations of root water uptake and comparison with sap flow measurements

The cumulative $ET_{pot}$, $T_{pot}$, and RWU simulated by the FJ and C models in the three plots of stony and silty soils during the whole measurement period and during the overlapping period of measurements in both soils are shown in Fig. 8. The higher cumulative $ET_{pot}$ in the stony plot than in the silty plots is simply due to the longer measurement period in the stony plot. The lower $ET_{pot}$ in the sheltered plot results from the lower net-radiation due to sheltering as compared to the neighboring unsheltered plots. The difference in cumulative $ET_{pot}$ between the stony and silty soils during the overlapping measurement period results from different $K_c$ values due to different time of ripening of the crop in the two soils (Table 1). The ratio $T_{pot}/ET_{pot}$ was considerably smaller in the stony soil than in the silty soil since the early crop development stage, when the crop canopy was not fully covering the soil and the LAI was low, was not covered by the measurement period in the silty soil. Differences in LAI also explain the smaller $T_{pot}/ET_{pot}$ ratio in the sheltered plot of the stony soil compared with the rainfed and irrigated plots of this soil and the larger $T_{pot}/ET_{pot}$ in the silty than in the stony soil during the overlapping measurement period. This illustrates that the potential water uptake by the wheat crop from the sheltered plot of the stony soil differs substantially from that of the other plots due to a different crop development and LAI. Only in the sheltered and rainfed plots of the stony soil, the simulated $T_{act}$ or RWU was reduced compared to the $T_{pot}$. In the silty plot, there was no reduction in simulated $T_{act}$ compared to $T_{pot}$ indicating that the calculated soil water supply in the root zone in the silty soil was sufficient for meeting the atmospheric demand.



Figure 9 shows potential and actual RWU simulated by the FJ and C models, and sap flow in the three plots of the stony soil and the silty soil from 23 May to 6 July 2014. When the measured sap flow was regressed against the simulated RWU by the two models, there was a good agreement between crop transpiration obtained from the sap flow measurements and model simulations with $r^2$ of 0.86 by the FJ model and 0.85 by the C model. But, there was a constant offset of 0.05 cm d$^{-1}$ between the sap flow measurements and the simulated RWU (Fig. 10a). The observed sap flow and the simulated $T_{act}$ were both higher in the silty than in the stony soil. In the silty soil, the sap flow measurements did not differ considerably between the different water treatments, which was consistent with the simulated $T_{act}$ that was equal to $T_{pot}$. For the stony soil, the measured sap flow differed between the different water treatments which was also consistent with the differences in simulated $T_{act}$.

There was, as far as we know, no similar comparison between sap flow and simulated RWU using field observations for wheat crop. Due to the "delicate anatomy of the walls of hollow wheat stems" (Langensiepen et al., 2014), it is challenging to install the sensors and measure the temperature variation of the thin wheat stalk with high time frequency for the field condition. Furthermore, spatial variation in environmental conditions that influence the sap flow in a single stem and variability in stem development lead to a considerable stem to stem variability in sap flow in which the average deviation from mean sap flow is quantified for the three different treatments shown in Fig.9 (Chabot et al., 2005; Zhang et al., 2014). The simulated RWU was based on a chain of models linked with assumptions and preset parameterizations starting from the calculation of the potential crop evapotranspiration using the empirical FAO56 approach, its split into soil evaporation and transpiration as a function of LAI, and its reduction to actual transpiration as a function of soil water potential. The overall good correlation between simulated RWU and sap flow measured transpiration therefore gives some confidence in the used approaches.

In order to unravel further the model's capability to calculate RWU in different soils and for different water treatments, we made plots of the ratios of the measured sap flow in the two soils versus the ratios of simulated RWU in the two soils for the different water treatments (Fig. 10b and c). Ratios were used to cancel out the temporal variations due to varying meteorological conditions. The good agreement between measured and simulated ratios for the irrigated plots, in which RWU was not influenced by water availability, indicates that the differences in potential transpiration rates between the two plots due to different crop development (ripening) and LAI were adequately represented in the models. There is no difference between the FJ and C models since RWU is completely defined as a boundary condition and not dependent on the soil water status in the irrigated plots, which was discussed by Cai et al. (2017). For the rainfed and sheltered plots, the correlation between the measured and simulated ratios is smaller. These ratios represent to what extent the simulated reduction of RWU in the stony soil due to reduced water availability is consistent with the measured reduction in sap flow relative to the simulated RWU and measured sap flow in the silty plots where there was no reduction in RWU. Of note is that simulations by the C model are more consistent with the sap flow measurements than the simulations by the FJ model. First, the FJ model predicts a larger reduction in RWU than the sap flow measurements suggest (see Fig. 9). Secondly, the ratios of the FJ model simulations vary less than the ratios of the sap flow measurements whereas the range of ratios of the C model simulations is more in agreement with the sap flow measurements. This indicates that the C model represents better than the FJ model how changing soil moisture and soil moisture distributions change the RWU. Furthermore, since the root hydraulic conductance in the C model depends on the root density, the model can reflect the impact of the differences in root density between the different water treatments on RWU. The FJ model did not possess this flexibility since only one set of water stress parameters was used for the different water treatments. Similar observations were made by Vandoorne et al. (2012) who optimized the water stress parameters of the FJ model for Chicory (*Cichorium intybus L.*) and found that the values of those parameters had to be adapted for different soil moisture conditions and different plant growth stages.





Sap flow per unit soil surface area was obtained by multiplying the average sap flow in the measured tillers with the number of tillers per unit soil surface area. Figure 11 shows the average sap flow per tiller and the sap flow per unit leaf area index. For the silty soil, the sap flow per tiller and sap flow per leaf area were very similar for the different water treatments. For the stony soil, the sap flow per tiller in the irrigated plot was similar to that in the silty soil until approximately 15 June. After that,

the sap flow per tiller reduced in the irrigated plot of the stony soil because of the reduction in leaf area (the sap flow per leaf area remained similar to that in the silty soil). Water stress limited the leaf development of wheat in both longevity and quantity (Khalid et al., 2016; Zhou et al., 2015). The sap flow per tiller in the rainfed plot of the stony soil became smaller than that in the irrigated plot or in the silty soil after 11 June but recovered for a short time period to same sap flow after the rainfall on 10 June. This recovery was also observed for the sap flow in the sheltered plot of the stony soil. But, the sap flow per tiller was

generally lower in this plot than in the other plots. This indicates that transpiration in this plot was reduced by both a reduced number of tillers and a lower flux per tiller. It is interesting to note that the sap flow per leaf area surface in the sheltered stony plot shortly increased to higher values than in other plots after the rainfall event on 10 June.

**3.4 Effects of root and shoot development on simulated transpiration**

The different root development in the two soils and for the different water treatments (Fig. 3) was related to a different

parameterization of the root hydraulic conductance (Fig. 7). The different shoot development and different LAI values (Fig. 2a) affected calculations of potential transpiration rates (Fig. 8) that were used as boundary conditions for RWU simulations. In order to demonstrate the impact of the plant development on the RWU simulation, we conducted two sets of simulations in which the plant parameters were prescribed by measurements done in another soil and/or water treatment. In the first set of simulations, we changed the root hydraulic conductance, $K_{rs}$. For the stony soil, $K_{rs}$ of all plots were rescaled by a factor of

1.78 which corresponds to the ratio of $K_{rs}$ in the sheltered plot of the silty soil in week 15 to $K_{rs}$ in the sheltered plot of the stony soil in the same week. This rescaling represents how RWU would change if the plants would not reduce the root hydraulic conductance in the stony soil. For the silty soil, $K_{rs}$ was scaled by a factor of 0.56, which is the inverse of the factor used to scale the root conductance in the stony plot. The rescaling for the silty plot represents how water uptake in the silty soil would be reduced if the root conductance was equal to that in the sheltered plot of the stony soil. For the stony plot, rescaling (i.e.

increasing) the root conductance increased the cumulative water uptake only by about 2 % in all plots (see Table 5). Increasing the root conductance therefore did not increase substantially the amount of water that could be extracted from the stony soil. For the silty soil, rescaling (i.e. decreasing) the root conductance in fact generated water stress and reduced the RWU by 9 %. Therefore, the root system with higher root densities and conductance in silty soil is apparently not 'over-dimensioned' whereas increasing the root conductance in the stony soil would hardly lead to more water uptake.

In a second set of simulations, we changed the calculated potential transpiration of the sheltered stony plot to that of the irrigated stony plot (Fig. 12). Only the stony soil was considered since the shoot and LAI development did not differ considerably between the different water treatments in the silty plot. Until 1 May, there was almost no difference in the LAI and $T_{pot}$ among different plots so that there was also no big effect on the simulated $T_{act}$. $T_{pot}$ in the irrigated plot started to

deviate from $T_{pot}$ in the sheltered plot from 1 May due to higher LAI in the irrigated plot (Fig. 2a). Increasing $T_{pot}$ in the sheltered plot did not affect the simulated RWU by the C model. In this model, the boundary condition switches to a constant pressure head boundary condition when stress occurs so that the simulated root water becomes independent of the potential transpiration rate.

Of interest is also the time at which water uptake starts to decrease and its effect on plant development. In the sheltered and rainfed stony plots, a slight reduction in RWU is simulated during April. This reduction in RWU was accompanied by only a slight decrease in LAI development compared to the irrigated plot (Fig. 2a). After mid of May, which is also the period when



RWU more strongly reduced, the LAI did not increase anymore in the sheltered plots whereas in the other plots of the stony soil, it reached its maximum at the beginning of June. In the silty soil, the maximum was reached at the beginning of July. The root system reached its full development, however, earlier than the time when the LAI reaches its maximum (Fig. 3). The root system development in the stony plot was much stronger reduced by the lower water availability in April than the LAI

development. Both leaves and roots showed reaction to environmental changes but this reaction was not simultaneous. Walter and Schurr (2005) reviewed studies of leaf and root growth of herbaceous plants and indicated that roots experienced more directly the effect of environmental factors (i.e. water stress, nutrient deficiency) compared with leaves. They also indicated that roots responded faster than leaves to the environmental conditions to optimize resource use efficiency.

**4 Conclusions**

The different crop development of winter wheat had consequences for the parameterization of RWU models. First, the different shoot development led to differences in boundary conditions such as the potential evapotranspiration ($K_c$ factor) and the potential transpiration (LAI). Differences in root development led to differences in root density distributions, root system conductivities, and RWU stress parameters. Water stress led to smaller root system conductances in the C model and a reduction of the RWU for less negative soil water potentials in the FJ model. Such a down-regulation of the root system conductance

due to drought stress has also been reported by Maurel et al. (2010), Trillo and Fernandez (2005), and Wang et al. (2013) for wheat, and Matsuo et al. (2009) for rice.

The C model, which is based on a physical description of the flow in the soil-root system, represented the effect of the differences in root system development on RWU directly since it relates the root system conductance to the root length. When

root parameters that were obtained from the sheltered stony plot were used to predict RWU in the silty soil, water stress was simulated in the silty soil. On the other hand, when root parameters obtained from the silty soil were used to simulate water uptake in the stony plot, the water uptake could only slightly be increased but the 'severity' of the water stress remained the same. This suggests that the root system that developed in the stony soil would be under-dimensioned for the silty soil and the opposite for the root system that developed in the silty soil.

The simulated differences in transpiration from the two different soils and the different water treatments could be confirmed by sap flow measurements. The physically-based C model predicted the ratios of the transpiration fluxes in the two soil types slightly better than the FJ model. Since the transpiration from the silty soil was close to $T_{pot}$, these ratios represented to what extent the transpiration was reduced due to reduced water availability in the stony soil.

This study illustrated that a combined dataset of root and shoot development, of soil water contents and soil water potentials, and of transpiration fluxes derived from sap flow measurements can be used to parameterize and validate RWU models. These models require inputs about root and shoot developments, which were observed to depend strongly on the environmental conditions. In how far the C model can improve prediction of RWU, transpiration and soil water stock depletion in widely

used crop models for different crops and climate conditions is subject of further investigations. Next to improving the description of the RWU, the C model also simulates the water potential in the root collar. In the current model formulation, the water potential in the collar is used as a control variable which is kept fixed when a critical threshold value is reached. We interpreted the reduction in transpiration when this threshold was reached as 'water stress'. However, we observed considerable reduction in aboveground biomass even when no reduction in transpiration was simulated (or observed with sap flow

measurements), e.g. in the sheltered and rainfed silty soil plots. Next to transpiration, stomatal opening, and carbon assimilation, plant growth is also linked to the hydraulic status of the shoot (Tardieu et al., 2014). Improving the prediction of the shoot





water potential, which is closely linked to the water potential in the root collar, is therefore also important to predict the reaction of plant growth to environmental conditions related to drought stress.

Concerning the observations of the root development using horizontally rhizotubes, it needs to be further investigated how
5   root counts along rhizotubes can be translated to root densities. Also, the reasons for the constant offset between the simulated transpiration and the sap flow measurements need to be further investigated.

*Competing interests.* The authors declare that they have no conflict of interest.

10  *Acknowledgements.* This study was financially supported by SFB/TR 32 (Transregional Collaborative Research Centre 32, funded by the Deutsche Forschungsgemeinschaft (DFG)). The meteorological data were obtained from the online database of the project TERENO. The rhizotron facility is part of the TERENO network of terrestrial observatories.





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





**Table 1.** Crop coefficients ($K_c$) of winter wheat in the stony (F1) and silty (F2) soils in different growing periods. $K_c$ was calculated according to Allen et al. (1998).

|  | Initial period 31.10.2013 – 27.02.2014 | Mid-season 08.05. – 27.06.2014 (F1) 08.05. – 09.07.2014 (F2) | Late stage 17.07.2014 (F1) 31.07.2014 (F2) |
|---|---|---|---|
| F1 | 0.93 | 1.26 | 0.27 |
| F2 | 0.93 | 1.26 | 0.29 |

**Table 2.** Parameters of soil hydraulic properties at the top- (0 – 30 cm) and subsoil (30 – 120 cm) of the stony (F1) and silty (F2) soils. $\theta_r$ and $\theta_s$ are residual and saturated soil water content, respectively. $\alpha$ and $n$ are curve-fitting parameters.

|  | $\theta_r$ cm³ cm⁻³ | $\theta_s$ cm³ cm⁻³ | $\alpha$ cm⁻¹ | $n$ |
|---|---|---|---|---|
| F1 topsoil | 0.0430 | 0.3256 | 0.0361 | 1.3860 |
| F1 subsoil | 0.0543 | 0.2286 | 0.0495 | 1.5340 |
| F2 topsoil | 0.1392 | 0.4089 | 0.0231 | 1.2920 |
| F2 subsoil | 0.1304 | 0.4119 | 0.0050 | 1.1920 |

**Table 3.** Tiller density (counted on 11 June 2014), crop biomass (including straw and grain which were measured after the harvest), ratio of leaf area index (LAI) to tiller density, and maximal root length in the three plots (P1: sheltered; P2: rainfed; P3: irrigated) of the stony (F1) and silty (F2) soils.

|  |  | P1 | P2 | P3 |
|---|---|---|---|---|
| Tiller density (m⁻²) | F1 | 228 | 310 | 370 |
|  | F2 | 346 | 380 | 390 |
| Biomass (kg m⁻²) | F1 | 0.2951 | 0.6719 | 0.7164 |
|  | F2 | 0.7164 | 1.0659 | 1.2097 |
| LAI/Tiller density (m²) | F1 | 0.0050 | 0.0067 | 0.0067 |
|  | F2 | 0.0075 | 0.0074 | 0.0075 |
| Maximal total root length (m m⁻²) | F1 | 2533.9 | 2941.9 | 3431.2 |
|  | F2 | 6787.4 | 7043.9 | 7024.1 |

**Table 4.** The saturated hydraulic conductivity ($K_s$), model shape parameter ($l$), critical pressure head in the Feddes water stress function ($h_{3l}$, $h_{3h}$), the compensation parameter ($\omega_c$), and the root system related parameters ($K_{rs}$ and $K_{comp}$) estimated by the Feddes-Jarvis (FJ) and Couvreur (C) models of the stony (F1) and silty (F2) soils and the corresponding objective function (OF) values.

| Site | Model | $K_{s1}$ (cm h⁻¹) | $l_1$ (-) | $K_{s2}$ (cm h⁻¹) | $l_2$ (-) | $h_{3l}$ (cm) | $h_{3h}$ (cm) | $\omega_c$ (-) | $K_{rs\_ini}$* (cm h⁻¹)† | $K_{comp\_ini}$* (cm h⁻¹) | OF |
|---|---|---|---|---|---|---|---|---|---|---|---|
| F1 | FJ | 0.663 (3.417)†† | 4.669 (1.470) | 1.581 (0.026) | 3.459 (-2.797) | -694 (-1172) | -238 (-648) | 0.95 (0.8) | - | - | 33.42 (41.79) |
|  | C | 0.426 (3.853) | 3.773 (1.472) | 1.556 (0.021) | 3.947 (-2.892) | - | - | - | 8.38*10⁻⁸ (8.77*10⁻⁸) | 2.71*10⁻⁸ (2.60*10⁻⁸) | 33.40 (40.97) |
| F2 | FJ | 0.450 | -1.358 | 0.144 | -3.165 | -747 | -279 | 0.95 | - | - | 31.93 |
|  | C | 0.417 | -2.219 | 0.623 | 1.379 | - | - | - | 5.99*10⁻⁸ | 3.32*10⁻⁹ | 35.90 |





† $K_{rs\_ini}^*$ and $K_{comp\_ini}^*$ are $K_{rs\_ini}$ and $K_{comp\_ini}$ normalized by root length per surface area. †† Parameters obtained using only measurements in the sheltered plot of the stony soil (Cai et al., 2017).

**Table 5.** Cumulative root water uptake simulated by the Couvreur model using root hydraulic conductance ($K_{rs}$) obtained from the silty soil (F2) for the stony soil (F1) and using $K_{rs}$ obtained from the stony soil for the silty soil. P1, P2, and P3 are the sheltered, rainfed, and irrigated plots. Values in parentheses are simulated RWU using the optimized parameters from measured SWC and SWP in the respective plots.

|    | P1 | P2 | P3 |
|----|-----|-----|-----|
| F1 | 13.55 (13.27) | 19.38 (19.01) | 25.40 (25.02) |
| F2 | 14.04 (15.36) | 17.40 (19.12) | 17.96 (19.65) |

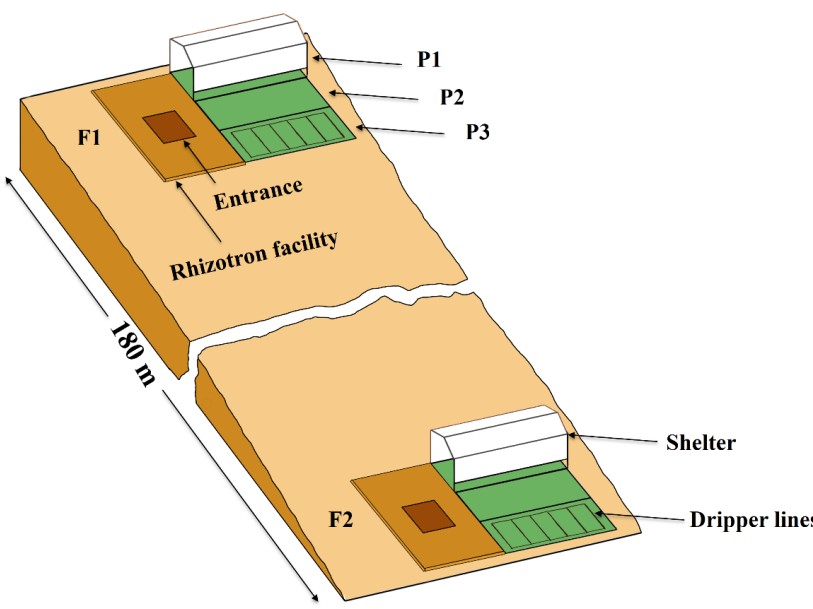

**Figure 1.** Sketch map of the location and the setup of the upper (F1, stony soil) and lower (F2, silty soil) rhizotron facilities. P1, P2, and P3: the sheltered, rainfed, and irrigated plots.



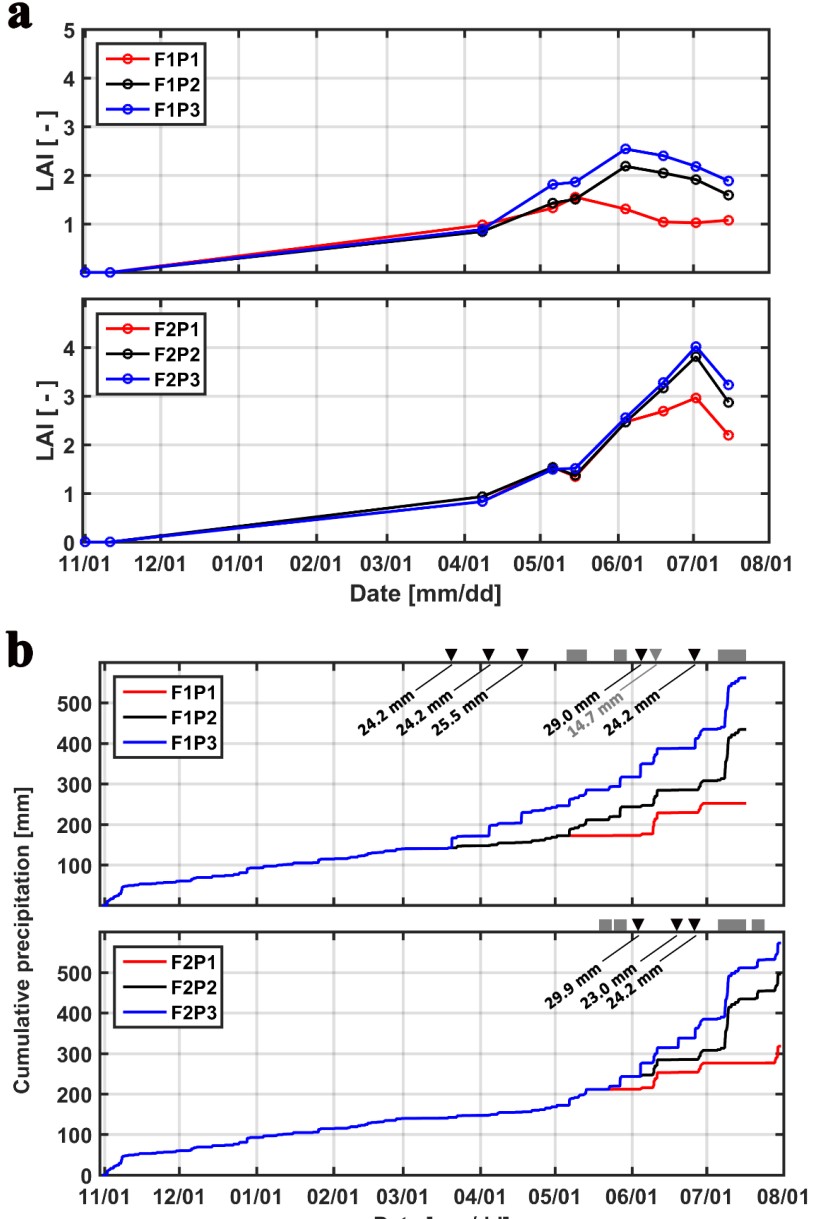

**Figure 2.** (**a**) Measured leaf area index (LAI) in the three plots (P1: sheltered; P2: rainfed; P3: irrigated) of the two facilities (F1: stony soil, F2: silty soil). (**b**) Cumulative precipitation and irrigation applied to the sheltered (P1), rainfed (P2), and irrigated (P3) plots of the stony (F1) and silty (F2) soils. ■: sheltered period, ▼: irrigation in P3, ▼: irrigation in P1.





**Figure 3.** Depth-time distribution of root length density (RLD) in in the three plots (P1: sheltered; P2: rainfed; P3: irrigated) of (**a**) the stony (F1) and (**b**) silty (F2) soils from 11 Feb. to 11 July 2014 and from 14 Mar. to 24 July 2014, respectively. No measurements in the gray grids.



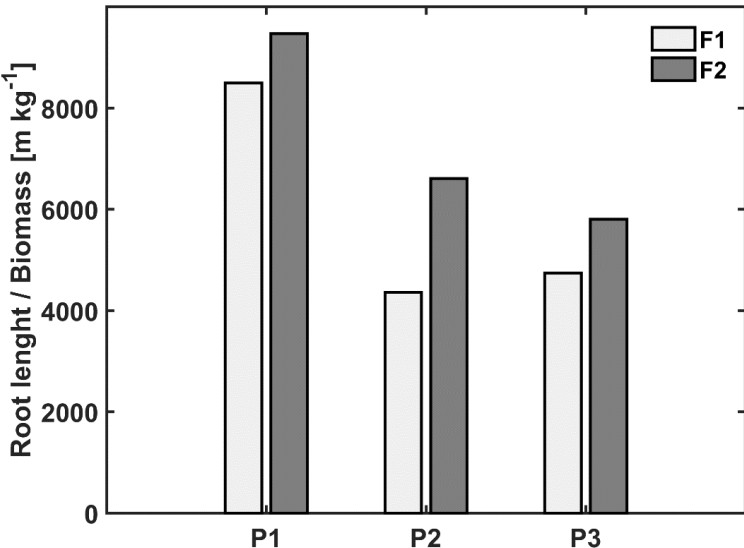

**Figure 4.** Ratio of total root length to biomass in the three plots (P1: sheltered; P2: rainfed; P3: irrigated) of the stony (F1) and silty (F2) soils.

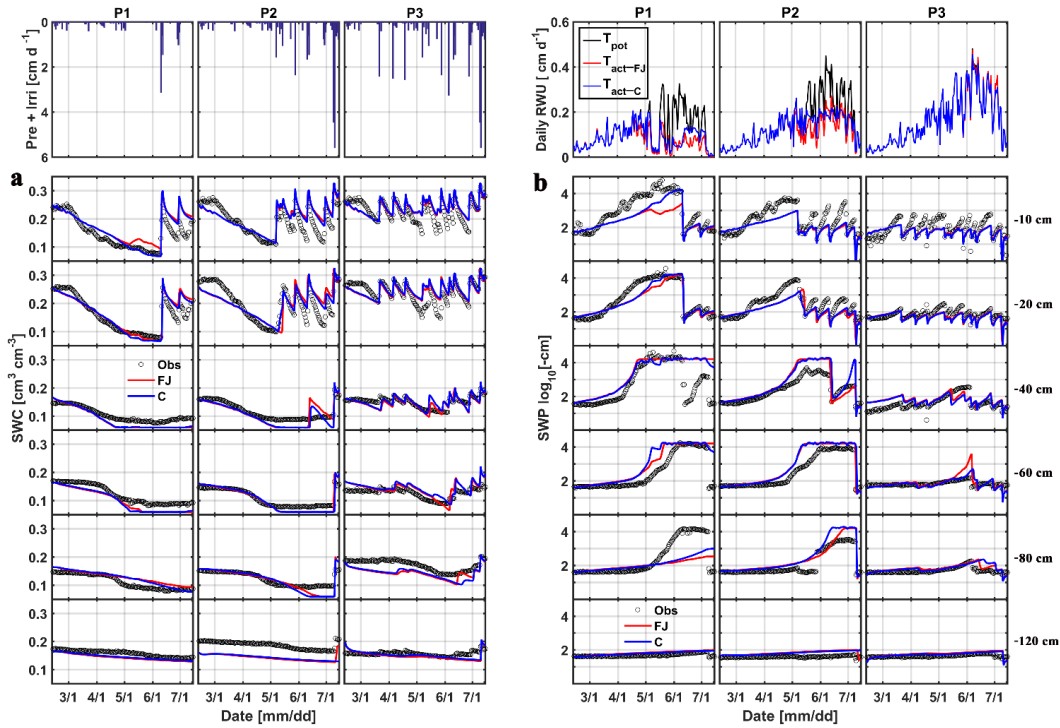

**Figure 5.** Comparison between observed (black) and simulated (**a**) soil water content (SWC) and (**b**) soil water pressure head (SWP) by the Feddes-Jarvis (FJ, blue) and Couvreur (C, red) models at six soil depths in the sheltered (P1), rainfed (P2), and irrigated (P3) plots of the stony soil (F1) from 11 Feb. to 14 July 2014. Time series of precipitation (Pre) and irrigation (Irri)



**Figure 6.** Same as Figure 5 but for silty soil from 22 May to 30 July 2014.




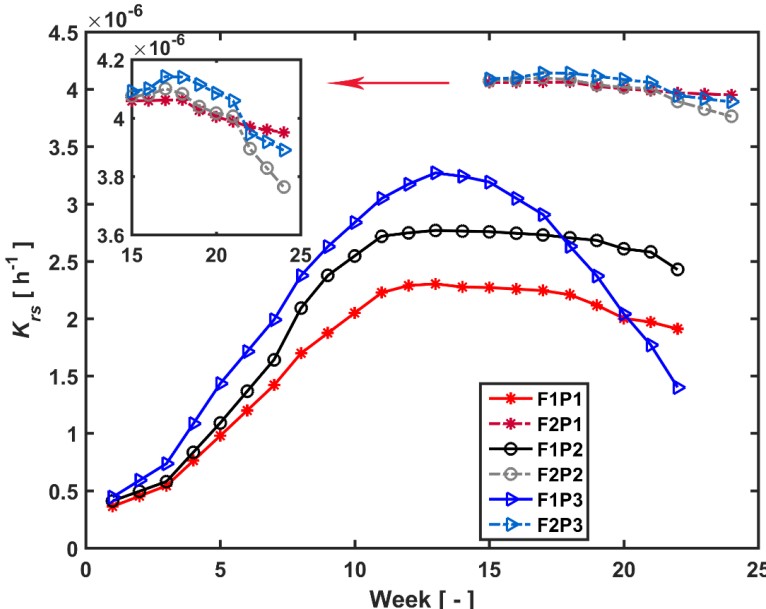

**Figure 7.** Estimated root hydraulic conductance ($K_{rs}$) in the three plots (P1: sheltered; P2: rainfed; P3: irrigated) of the stony (F1) and silty (F2) soils during the measurement period (F1: from 11 Feb. to 11 July 2014, F2: from 23 May to 24 July 2014).

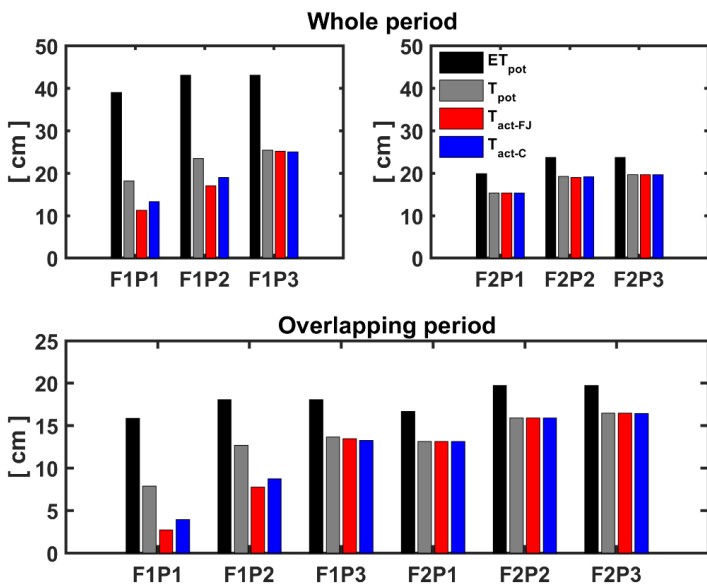

**Figure 8.** Potential evapotranspiration ($ET_{pot}$), potential transpiration ($T_{pot}$), and actual transpiration ($T_{act}$ = RWU) estimated by the Feddes-Jarvis (FJ) and Couvreur (C) models in the three plots (P1: sheltered; P2: rainfed; P3: irrigated) of the stony (F1) and silty (F2) soils in the whole period (F1, from 11 Feb. to 14 July 2014; F2, from 22 May to 30 July 2014) and in the overlapping period (from 22 May to 14 July 2014).



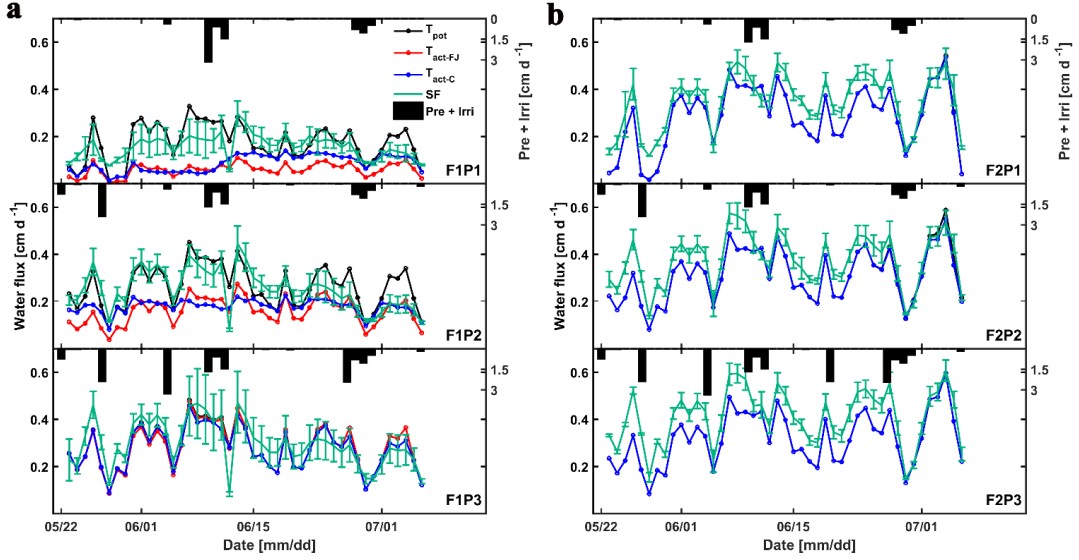

**Figure 9.** Daily cumulative potential transpiration ($T_{pot}$), root water uptake ($T_{act}$ = RWU) simulated by the Feddes-Jarvis (FJ) and Couvreur (C) models, and sap flow (SF) in the three plots (P1: sheltered; P2: rainfed; P3: irrigated) of (**a**) the stony (F1) and (**b**) silty soils (F2) from 23 May to 6 July 2014. Pre: precipitation, Irri: irrigation.



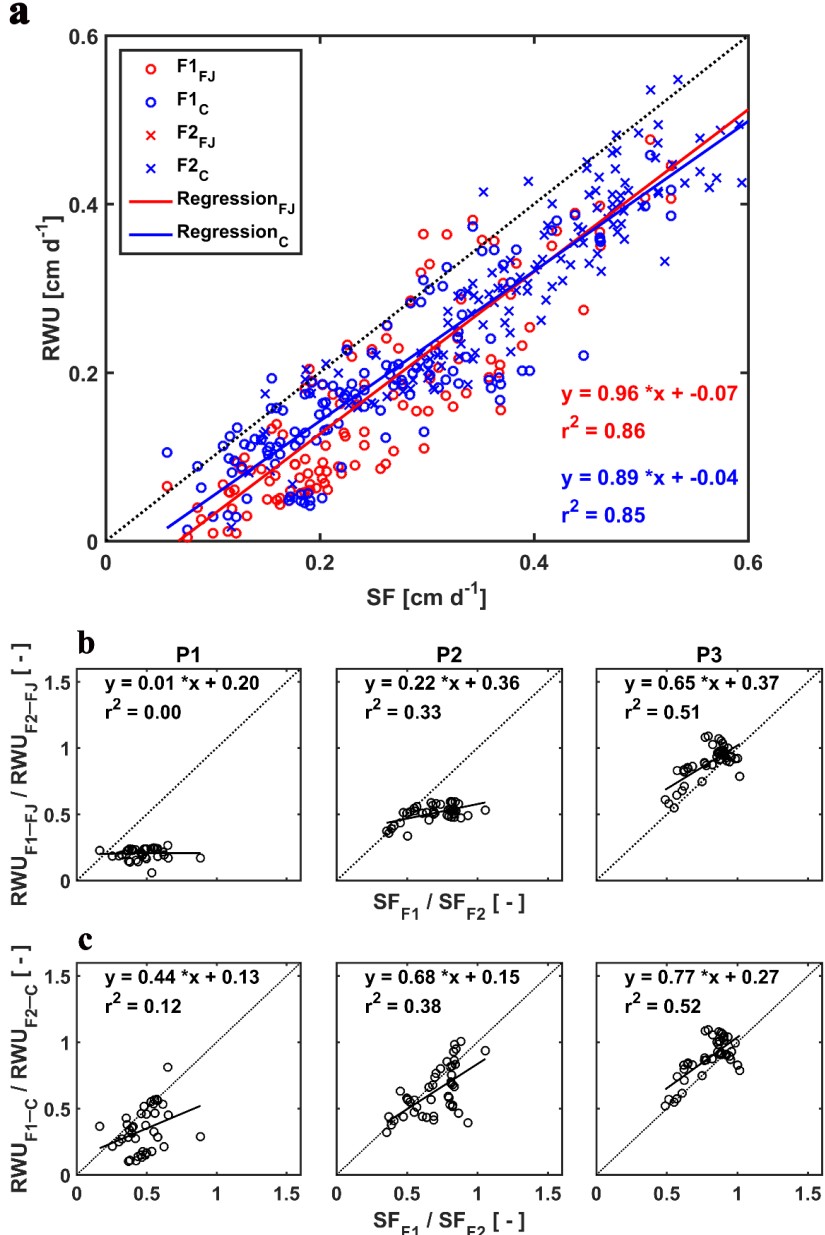

**Figure 10.** Correlation (**a**) between sap flow (SF) and root water uptake (RWU) simulated by the Feddes-Jarvis (FJ) and Couvreur (C) models of the stony (F1) and silty soil (F2). Relation between the ratio of the RWU in the stony to the RWU in the silty soil estimated by the FJ (**b**) and C (**c**) models in the three plots (P1: sheltered; P2: rainfed; P3: irrigated) versus the ratio of sap flow in the stony soil to that in the silty soil.



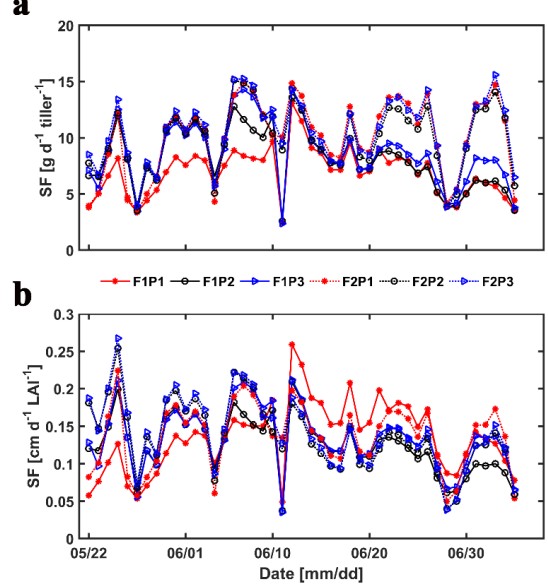

**Figure 11.** Measured sap flow (SF) per tiller (**a**) and per unit leaf area index (LAI) (**b**) in the three plots (P1: sheltered; P2: rainfed; P3: irrigated) of the stony (F1) and silty (F2) soils.

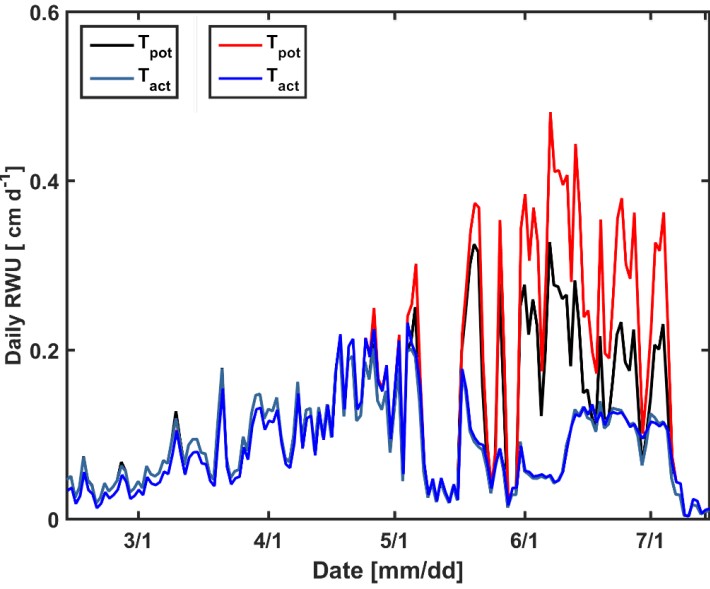

**Figure 12.** Daily potential ($T_{pot}$) and actual transpiration ($T_{act}$ = RWU) estimated by the Couvreur model using the measured leaf area index (LAI) in the sheltered plot (black and cyan) of the stony soil and the LAI in the irrigated plot of the stony soil (red and blue).