# Peer review of "Root growth, water uptake, and sap flow of winter wheat in response to different soil water availability"

_Hydrology and Earth System Sciences, 2017_

## Referee Comment (RC1) · Anonymous Referee #1 · 4 Jan 2018

Review of Root growth, water uptake, and sap flow in winter wheat in response to different soil water availability

Cai et al.

In this manuscript, the authors compare and assess the ability of two models of soil-moisture dynamics to represent root-water uptake for a detailed field experiment with two different soils and three different watering schemes. The paper is well-written and detailed. I have only minor suggestions that I offer in the spirit of improving clarity and message.

1. Intent In the opening sentence of the abstract, the authors state "How much and

where water is taken up by roots from the soil profile are important questions that need to be answered to close the soil water balance equation and to describe water fluxes in the soil-plant-atmosphere continuum." While I do not wish to quibble with the importance of these questions, I would offer that a detailed description of from where in the root zone water is removed is not necessarily needed to close the soil-water balance and describe water fluxes. Indeed, there are simple models of the root-zone, some without vertical resolution, that "adequately" represent overall water uptake and the soil-water balance with varying degress of precision.

That said, I agree that there is a need or opportunity for more detailed models that can represent spatial variations in water uptake – both for description (i.e., uncovering and representing the processes at work within the vadose zone) and for prediction. I suggest that the authors offer a stronger articulation of intent for their work – is it that they care most about overall water balance, and these two models offer the appropriate degree of flexibility and/or complexity? Or, is it that they care most about the vertical variations in water uptake? (i.e., is it figures 5 and 6 or figures 8 and 9 that are most important)? Why these two models versus others that could be employed? This is not a major point, but I think the results will be more impactful if the authors can make clearer how the results are extensible to other situations and when such details are warranted (and when not). Why these models and what are the modeling goals?

I finished the paper impressed with the work and conscientious detail that had gone into the field and modeling experiments, but also without a clear sense of how I would incorporate these results and conclusions into future work – a clearer statement to that effect would help.

2. Big-picture site context The field experiment that is being described and modeled in this work is quite detailed. I sometimes found myself losing the forest for the trees. I recommend embellishing section 2.1 to provide the big-picture context for the reader.

First, I recommend providing insight to the climate/aridity. Perhaps simply stating the

overall ratio of precipitation to reference evapotranspiration for the duration of the study. Obviously, the treatments modulate the available water, but providing a starting point for the aridity (and whether the climate is fundamentally arid or humid) would help.

Additionally, differences between the two soils are many and nuanced. At the same time, offering a statement in section 2.1 regarding the plant-available water content (i.e., the difference in water content at field capacity [however defined] and at the wilting point) for both soils would help the reader understand some of the fundamental differences. Are they similar? Very different? What about hydraulic conductivity? What, from the authors' perspective are the 1 or 2 key differences between the two soils, as they relate to the objectives of the study?

3. Figure 8 Figure 8 is quite helpful. However, it is a bit misleading/confusing. I recommend the following: - label the sub-figures with the dates themselves rather than the less clear terms such as "whole period" and "overlapping period." This will also help the reader understand the differences in potential ET, since the stony soil data start much earlier, when LAI is low. - add a bar for the sapflux estimates of water uptake to provide additional context, especially since some of the low values for the stony soil seem to be partly due to error (see figure 9) - finally, while it may begin to get a bit crowded, I also recommend including a bar for precipitation/irrigation. Such a bar would really help contextualize the similarity or differences among the actual and potential transpiration.

4. Figure 9 Why is it that the silty soil plots have just two lines (sap flux and Couvreur model)?

5. Figure 10 I understand how taking the ratio of transpiration eliminates variations in potential ET – makes sense. Using these ratios, however, also masks the absolute error in the estimates of ET from the stony soils (see Figure 9). That's fine, but it should be more appropriately acknowledged. For example, on line 34 of page 11, the authors articulate that the FJ model underestimates sap flux. However, the same is true of the

[Figure]

C model, and this statement seems to be stretching the superiority of the C model over the FJ model. (The second statement about better representing the variability between soils seems to be the primary difference.)

Also, I do not see any red x's in Figure 10a (for FJ model in silty soil).

Overall, I found the paper to be thoughtful, detailed, and well-written

---

## Referee Comment (RC2) · Anonymous Referee #2 · 16 Jan 2018

This paper compares the performance of two root water uptake models against a field dataset of soil water contents/potentials and sap flow measured in two contrasting soil types for three different watering regimes. The dataset is comprehensive, the model application has been performed carefully, and the results are also very interesting. The paper should make a valuable contribution to the literature on this important topic.

One concern I have is that the methods are not fully described. Firstly, the water uptake models themselves are not well explained. The equations are given, but the readers are given no indication of how they have been derived. The authors should explain that although it is physics-based, the C model is an approximate solution to

a 3D root architecture model that does involve some assumptions and simplifications. For completeness, these should be stated. For the empirical FJ model, the authors should give some background information on what the main functions and parameters in the models are supposed to reflect (there is actually some physical basis to the model). Similarly, although a detailed description is not necessary, the authors should at least mention the basic principles of the method they used to calibrate the model parameters.

The authors emphasize that one important advantage of the physics-based C model is that it accounts for the effects of total root conductance (or root length) on uptake, whereas the empirical (phenomenological) FJ model only considers a relative root distribution. This is certainly true of the way the FJ model was originally formulated and is still mostly used. However, I think the authors should mention in the paper that the analysis in Jarvis (2011) shows that the compensation parameter omega_c in the FJ model should be dependent on the ratio of the potential transpiration rate to the total root length/conductance. From this point of view, it would have been better to calibrate omega_c separately for each combination of soil type and watering treatment. The derived values could then have been compared with the measured LAI/root length ratios. With a smaller LAI/root length ratio, the covered treatment (especially in the stony soil) should have smaller omega_c values. This could also have given better simulations of the sap flow data. This lumping of the treatments might also explain why the calibration of the FJ model seemed to suffer from poorly defined parameters (equifinality) and also why the overall calibrated omega_c values were 0.95 at both sites, which implies that virtually no compensation occurred. This result should also be discussed in the paper in light of the above points, because otherwise it might seem very surprising to the reader given the drought conditions that were induced in the covered treatment. Of course, ideally, model parameters should be constant! But in this case, I think it could help understanding to explore and discuss why omega_c might not be constant.

Specific (minor) comments

Abstract, Line 15 (and Page 10, lines 4-14): this result is only shown in the supplementary. If it is important enough to mention in the abstract, then it should be shown in the paper itself.

Page 4, line 10: how close? Please give the exact distance.

Page 4, line 18: Are these rainfall totals, not precipitation? You need to be careful about the choice of words here, because of the irrigation supplied to some plots.

Page 6, line 28: some brief details of the method are needed here.

Page 7, lines 25-32: you could also discuss the effects of water treatment on LAI here. LAI may be more directly related to potential transpiration than above-ground biomass?

Page 8, line 22: "above-ground"

Page 8, lines 30-31: better to replace "stimulated" by "restricted" and swop "silty" and "stony"

Figure 10: perhaps this should be split into two figures?

Reference

Jarvis, N.J. 2011. Simple physics-based models of compensatory plant water uptake: concepts and eco-hydrological consequences. Hydrology and Earth System Sciences, 15, 3431-3446.

———————————

---

## Editor Comment (EC1) · N. Romano (Editor) · 21 Jan 2018

Dear Authors, In view of the comments received so far, I suggest you should start posting some preliminary responses form your side so as to feed the discussion step of the journal.

---

## Short Comment (SC1) · 22 Jan 2018

Dear Prof. Romano,

Thanks for your kind reminder! I am currently preparing for my PhD defense this week. I will start to work on the reply to all the reviewers after the defense.

Kind regards, Gaochao Cai
* * *

---

## Author Comment (AC1) · 9 Mar 2018

In this manuscript, the authors compare and assess the ability of two models of soil moisture dynamics to represent root-water uptake for a detailed field experiment with two different soils and three different watering schemes. The paper is well-written and detailed. I have only minor suggestions that I offer in the spirit of improving clarity and message.

1. Intent In the opening sentence of the abstract, the authors state "How much an where water is taken up by roots from the soil profile are important questions that need to be answered to close the soil water balance equation and to describe water

fluxes in the soil-plant-atmosphere continuum." While I do not wish to quibble with the importance of these questions, I would offer that a detailed description of from where in the root zone water is removed is not necessarily needed to close the soil-water balance and describe water fluxes. Indeed, there are simple models of the root-zone, some without vertical resolution, that "adequately" represent overall water uptake and the soil-water balance with varying degress of precision.

Reply: the first sentence of the abstract was changed accordingly (line1 to line 2) on Page 1.

That said, I agree that there is a need or opportunity for more detailed models that can represent spatial variations in water uptake – both for description (i.e., uncovering and representing the processes at work within the vadose zone) and for prediction. I suggest that the authors offer a stronger articulation of intent for their work – is it that they care most about overall water balance, and these two models offer the appropriate degree of flexibility and/or complexity? Or, is it that they care most about the vertical variations in water uptake? (i.e., is it figures 5 and 6 or figures 8 and 9 that are most important)? Why these two models versus others that could be employed? This is not a major point, but I think the results will be more impactful if the authors can make clearer how the results are extensible to other situations and when such details are warranted (and when not). Why these models and what are the modeling goals?

Reply: we changed the sentences in the last but second paragraph of the introduction part on Page 3 to explain why we used these models (line24 to line 27). Actually we mentioned the advantages of using the Feddes-Jarvis model and used the second and third paragraphs on Page 2 to explain why we used the C model. Given the data that we have, we put the focus on the total uptake. We did not focus on uptake depths since that would require the use of isotope tracer data or measuring sap flow in the roots.

I finished the paper impressed with the work and conscientious detail that had gone into the field and modeling experiments, but also without a clear sense of how I would

incorporate these results and conclusions into future work – a clearer statement to that effect would help.

Reply: thanks for the suggestion! We added some sentences in the second and fourth paragraphs of the conclusions part on Page 14 (line 24 to line 28) and Page 15 (line 5 to line 8) to state the suggestions in future work (the fourth paragraph stated as well) based on what we obtained in this study.

2. Big-picture site context The field experiment that is being described and modeled in this work is quite detailed. I sometimes found myself losing the forest for the trees. I recommend embellishing section 2.1 to provide the big-picture context for the reader.

First, I recommend providing insight to the climate/aridity. Perhaps simply stating the overall ratio of precipitation to reference evapotranspiration for the duration of the study. Obviously, the treatments modulate the available water, but providing a starting point for the aridity (and whether the climate is fundamentally arid or humid) would help.

Reply: it was added accordingly at the end of the last paragraph of section 2.1 on Page 4 (line 23 to line 26).

Additionally, differences between the two soils are many and nuanced. At the same time, offering a statement in section 2.1 regarding the plant-available water content (i.e., the difference in water content at field capacity [however defined] and at the wilting point) for both soils would help the reader understand some of the fundamental differences. Are they similar? Very different? What about hydraulic conductivity? What, from the authors' perspective are the 1 or 2 key differences between the two soils, as they relate to the objectives of the study?

Reply: thanks for the suggestions! The information of plant-available water content was added in the first paragraph of section 2.1 on Page 4 (line 4 to line 7).

3. Figure 8 Figure 8 is quite helpful. However, it is a bit misleading/confusing. I recommend the following: - label the sub-figures with the dates themselves rather than

the less clear terms such as "whole period" and "overlapping period." This will also help the reader understand the differences in potential ET, since the stony soil data start much earlier, when LAI is low. - add a bar for the sapflux estimates of water uptake to provide additional context, especially since some of the low values for the stony soil seem to be partly due to error (see figure 9) - finally, while it may begin to get a bit crowded, I also recommend including a bar for precipitation/irrigation. Such a bar would really help contextualize the similarity or differences among the actual and potential transpiration.

Reply: thanks for your suggestions! - The dates were added in Figure 9 (new order). - We do not to include sap flow in this figure for two reasons: first, we did not focus on the difference in absolute total values between simulated RWU and measured sap flow; second, the sap flow was mentioned later and the measurement period of sap flow was different from the overlapped period of the estimated RWU between the two soils. - The precipitation and irrigation bars were added in Figure 9 (new order).

4. Figure 9 Why is it that the silty soil plots have just two lines (sap flux and Couvreur model)?

Reply: there are four lines in Fig. 10a and 10b (new order). No reduction of water uptake was simulated in the silty soil so that the values of Tact simulated by the FJ and C model were equal to Tpot. The lines overlapped. A sentence was added in the caption of Fig. 10 to clarify this issue.

5. Figure 10 I understand how taking the ratio of transpiration eliminates variations in potential ET – makes sense. Using these ratios, however, also masks the absolute error in the estimates of ET from the stony soils (see Figure 9). That's fine, but it should be more appropriately acknowledged. For example, on line 34 of page 11, the authors articulate that the FJ model underestimates sap flux. However, the same is true of the C model, and this statement seems to be stretching the superiority of the C model over the FJ model. (The second statement about better representing the variability between

soils seems to be the primary difference.)

Reply: this is correct. In order to avoid confusion, we skipped the sentence in line 33 on Page 12. The absolute difference between the sap flow measurements and the simulated root water uptake is discussed with figures 10 and 11 and we did not have to pick up that discussion again here.

Also, I do not see any red x's in Figure 10a (for FJ model in silty soil).

Reply: the same answer to question 4. The red x's and red circles overlapped since no water stress was simulated in the silty soil.

Please also note the supplement to this comment:
https://www.hydrol-earth-syst-sci-discuss.net/hess-2017-711/hess-2017-711-AC1-supplement.pdf

―――――――――――――――――

---

## Author Comment (AC2) · 9 Mar 2018

This paper compares the performance of two root water uptake models against a field dataset of soil water contents/potentials and sap flow measured in two contrasting soil types for three different watering regimes. The dataset is comprehensive, the model application has been performed carefully, and the results are also very interesting. The paper should make a valuable contribution to the literature on this important topic.

One concern I have is that the methods are not fully described. Firstly, the water uptake models themselves are not well explained. The equations are given, but the readers are given no indication of how they have been derived. The authors should

explain that although it is physics-based, the C model is an approximate solution to a 3D root architecture model that does involve some assumptions and simplifications. For completeness, these should be stated. For the empirical FJ model, the authors should give some background information on what the main functions and parameters in the models are supposed to reflect (there is actually some physical basis to the model).

Reply: we have the description of the FJ model in the introduction part (first paragraph of Page 2). As for the simplifications and assumptions, we added a paragraph after Eq. 4 on Page 5 (line 31 to line 41) and Page 6 (line 1 to line 4).

Similarly, although a detailed description is not necessary, the authors should at least mention the basic principles of the method they used to calibrate the model parameters.

Reply: a brief description of the method we used was added in the second paragraph on Page 7 (line 13 to line 15).

The authors emphasize that one important advantage of the physics-based C model is that it accounts for the effects of total root conductance (or root length) on uptake, whereas the empirical (phenomenological) FJ model only considers a relative root distribution. This is certainly true of the way the FJ model was originally formulated and is still mostly used. However, I think the authors should mention in the paper that the analysis in Jarvis (2011) shows that the compensation parameter omega_c in the FJ model should be dependent on the ratio of the potential transpiration rate to the total root length/conductance. From this point of view, it would have been better to calibrate omega_c separately for each combination of soil type and watering treatment. The derived values could then have been compared with the measured LAI/root length ratios. With a smaller LAI/root length ratio, the covered treatment (especially in the stony soil) should have smaller omega_c values. This could also have given better simulations of the sap flow data. This lumping of the treatments might also explain why the calibration of the FJ model seemed to suffer from poorly defined parameters (equifinality) and also

why the overall calibrated omega_c values were 0.95 at both sites, which implies that virtually no compensation occurred.

Reply: thanks for the suggestions! An additional paragraph was added to discuss the relation between the value of omega_c and the ratio of total root length to LAI on Page 10 (line 20 to line 31).

This result should also be discussed in the paper in light of the above points, because otherwise it might seem very surprising to the reader given the drought conditions that were induced in the covered treatment. Of course, ideally, model parameters should be constant! But in this case, I think it could help understanding to explore and discuss why omega_c might not be constant.

Reply: here we disagree with the statement that model parameters need to be constant. If the system properties change, the parameters that represent these properties need to change as well.

Specific (minor) comments Abstract, Line 15 (and Page 10, lines 4-14): this result is only shown in the supplementary. If it is important enough to mention in the abstract, then it should be shown in the paper itself.

Reply: thanks for the suggestion. The figure was put in the paper as Figure 8.

Page 4, line 10: how close? Please give the exact distance.

Reply: it was added in the second paragraph of section 2.1 on Page 4 (line 13 to line 14).

Page 4, line 18: Are these rainfall totals, not precipitation? You need to be careful about the choice of words here, because of the irrigation supplied to some plots.

Reply: they were the total amount of rainfall in the two soils. It was changed accordingly in the last paragraph of section 2.1 on Page 4 (line 21).

Page 6, line 28: some brief details of the method are needed here.

Reply: the method was briefly described in the first paragraph on Page 7 (line 13 to line 15).

Page 7, lines 25-32: you could also discuss the effects of water treatment on LAI here. LAI may be more directly related to potential transpiration than above-ground biomass?

Reply: thanks for the suggestion! The discussion of the effects of water treatment on LAI was added in the first paragraph of section 3.1 on Page 8 (line 14 to line 20).

Page 8, line 22: "above-ground"

Reply: it was replaced accordingly in line 9 on Page 9 (line 9).

Page 8, lines 30-31: better to replace "stimulated" by "restricted" and swop "silty" and "stony"

Reply: thanks for the suggestion! They are changed accordingly in line 17 and 18 on Page 9.

Figure 10: perhaps this should be split into two figures?

Reply: it was split into two figures (Figure 11 and 12) accordingly.

Please also note the supplement to this comment:
https://www.hydrol-earth-syst-sci-discuss.net/hess-2017-711/hess-2017-711-AC2-supplement.pdf